# Exploring *RAB11A* Pathway to Hinder Chronic Myeloid Leukemia-Induced Angiogenesis In Vivo

**DOI:** 10.3390/pharmaceutics15030742

**Published:** 2023-02-23

**Authors:** Catarina Roma-Rodrigues, Alexandra R. Fernandes, Pedro V. Baptista

**Affiliations:** 1Associate Laboratory i4HB—Institute for Health and Bioeconomy, NOVA School of Science and Technology, NOVA University Lisbon, 2829-516 Caparica, Portugal; 2UCIBIO—Applied Molecular Biosciences Unit, Department of Life Sciences, NOVA School of Science and Technology, NOVA University Lisbon, 2829-516 Caparica, Portugal

**Keywords:** chronic myeloid leukemia, exosomes, gold nanoparticles, small rab GTPase Rab11a

## Abstract

Neoangiogenesis is generally correlated with poor prognosis, due to the promotion of cancer cell growth, invasion and metastasis. The progression of chronic myeloid leukemia (CML) is frequently associated with an increased vascular density in bone marrow. From a molecular point of view, the small GTP-binding protein Rab11a, involved in the endosomal slow recycling pathway, has been shown to play a crucial role for the neoangiogenic process at the bone marrow of CML patients, by controlling the secretion of exosomes by CML cells, and by regulating the recycling of vascular endothelial factor receptors. The angiogenic potential of exosomes secreted by the CML cell line K562 has been previously observed using the chorioallantoic membrane (CAM) model. Herein, gold nanoparticles (AuNPs) were functionalized with an anti-*RAB11A* oligonucleotide (AuNP@RAB11A) to downregulate *RAB11A* mRNA in K562 cell line which showed a 40% silencing of the mRNA after 6 h and 14% silencing of the protein after 12 h. Then, using the in vivo CAM model, these exosomes secreted by AuNP@RAB11A incubated K562 did not present the angiogenic potential of those secreted from untreated K562 cells. These results demonstrate the relevance of Rab11 for the neoangiogenesis mediated by tumor exosomes, whose deleterious effect may be counteracted via targeted silencing of these crucial genes; thus, decreasing the number of pro-tumoral exosomes at the tumor microenvironment.

## 1. Introduction

Chronic myeloid leukemia (CML) is caused by a translocation between chromosomes 9 and 22, generating the aberrant Philadelphia chromosome [1]. The t (9;22) (q34;q11) translocation occurs between the Abelson murine leukemia (ABL1) and the Breakpoint Cluster region (BCR) genes resulting in the BCR-ABL1 gene that encodes a tyrosine kinase with constitutive enhanced activity, rendering uncontrolled proliferation to myeloid cells [2,3,4]. While the first line therapeutics with tyrosine kinase inhibitors (TKIs) has proven effective for most patients, acquired resistance due to TKIs commonly used in the clinics, such as imatinib or dasatinib, is often observed [2,5]. CML progression has also been correlated with increased angiogenesis at the bone marrow, providing the tumor microenvironment (TME) with a high profusion of capillaries [6,7]. The caliber and density of these blood vessels in bone marrow dictate the disease prognosis [6]. As such, tackling this neoangiogenesis is critical to effectively disrupt cancer progression. Several therapeutic strategies targeting the vascular endothelial factor receptor (VEGFR) have been effective in slowing down the progression of the disease [8,9,10]. Additionally, the role of CML-derived exosomes, i.e., endosomal derived nanovesicles involved in cell–cell communications [11,12,13], in triggering neoangiogenesis has been critical to understand the observed increase to blood vessel density in the bone marrow of CML patients [12,13,14,15,16,17].

The small Rab GTPase Rab11, involved in the endosomal slow recycling towards the plasma membrane [18], seems to play a pivotal role in neo-angiogenesis [19,20,21,22]. The secretion of VEGFR-1 and VEGFR-2 from recycling endosomes towards the plasma membrane is dependent of Rab11 vesicles [20,21,23]. When cells are not stimulated, VEGFR-2 is mainly localized in recycling endosomes associated with Rab5, involved in the initial internalization of proteins from the plasma membrane to early endosomes, and Rab4, involved in the fast-recycling pathway [20,24]. Simultaneous expression of VEGFR-2 and neuropilin-1 (NRP-1) result in proteins trafficking into the plasmatic membranes via Rab11 vesicles [20,23]. Moreover, it was described that Rab11 is also involved in the formation of angiogenic sprouts, by binding to phosphorylated vascular endothelial cadherin [22], and in the recycling of α5-integrin-p-FAK complexes involved in the assembling of adhesion sites in endothelial cells [25]. Additionally, in the CML cell line K562, exosomes secretion is mediated by Rab11 as a disposal way to regulate cellular levels of specific components [26,27]. The secretion of exosomes was shown to decrease when K562 cells had been transfected with a Rab11 GTPase deficient mutant, and a Rab11 GTP-binding deficient mutant [28]. Zhao et al. [19] also observed a decrease of exosome secretion by colorectal cancer cell line HCT116 after treatment with low doses of Apatinib, which was further observed that was due to regulation of multivesicular bodies biogenesis, transport and fusion, by regulation of LAMP2, VAMP2, Snap23 and Rab11.

Gold nanoparticles (AuNPs) have become significant players in the field of nano-medicine [29,30,31,32,33,34,35,36,37,38,39]. Due to their simple manufacture, high surface area, and distinctive physical and chemical properties, these nanoparticles are suited for use in a wide range of applications, including diagnosis, therapy, and imaging of various diseases [29,30,31,32,33,34,35,36,37,38,39]. The surface plasmon resonance (SPR) by itself, one of the physical characteristics that confers exceptional light-to-heat conversion efficiencies, contributes to the AuNPs use as photothermal agents [31,32,33,34,36,37,38]. Moreover, a wide range of biomolecules, including targeting and silencing moieties, dyes, or chemotherapeutic medicines, can easily be functionalized onto the surface of AuNPs, enabling the use of these particles for therapeutic, imaging, targeting applications, and combined therapies [30,31,32,33,34,35,36,37,38,39]. 

The chorioallantoic membrane (CAM) of chicken embryos has long been used as a highly accessible in vivo model for observation and manipulation of the angiogenic process [40]. In fact, the anti-angiogenic potential of peptide-coated gold nanoparticles (AuNP) [41], whose anti-angiogenic potential was improved with phototherapy [42], have been elegantly demonstrated using the CAM model. Moreover, we have previously shown the capability of exosomes derived from K562 leukemic cells to induce neo-angiogenesis in a VEGF dependent way, whose angiogenic effect could be counteracted with the anti-angiogenic peptide functionalized AuNPs [17].

Herein, we used AuNPs functionalized with an antisense oligonucleotide targeting *RAB11A* mRNA (AuNP@RAB11A) to examine the effect of Rab11a silencing on the in vivo angiogenic potential of exosomes secreted by the leukemic K562 cells. The silencing of *RAB11A* led to a decreased expression of Rab11a protein and consequent diminished secretion of exosomes. These “silenced” exosomes, secreted by AuNP@RAB11A incubated K562 cells, were not capable of inducing neoangiogenesis, as observed for the leukemic exosomes from standard K562 [17].

## 2. Materials and Methods

### 2.1. Preparation and Characterization of Gold Nanoconjugates

The preparation of citrate caped AuNPs, functionalization with PEG and with anti-*RAB11A* oligonucleotide followed protocols optimized in our group and described in [28]. Briefly, 25 mL of 38.8 mM sodium citrate was added to a 250 mL boiling solution of 1mM HAuCl_4_ (Sigma-Aldrich, St. Louis, MO, USA) and protected from light. After boiling for 30 min, the solution was cooled to room temperature (RT) and filtered in 0.2 μm filter syringe. Afterwards, 10 nM citrate caped AuNPs were mixed with 3 μg/mL of O-(2-Mercaptoethyl)-O’-methyl-hexa (ethylene glycol) (poly (ethylene glycol, PEG, Sigma-Aldrich) and 0.028% (*w*/*v*) of Sodium Dodecyl Sulfate (SDS, Sigma-Aldrich), and incubated overnight at room temperature. Resultant pegylated AuNPs (AuNP@PEG) were washed with deionized water through centrifugations at 12,000× *g* for 1 h. The AuNP@RAB11A nanoconjugates were prepared by mixing AuNP@PEG with an antisense hairpin oligonucleotide targeting *RAB11A* mRNA; 5′-GCTATGA TCG AGA CAG GAG ATT ACT CTT TCATAGC-3′, constructed based on the siRNA proposed by Lipinsky et al. [43] and palindrome (underlined) used for shDNA construction in Oliveira et al. [44]. The accomplishment of AuNPs functionalization was confirmed by UV/Vis spectroscopy and DLS. The antisense hairpin was designed to hybridize with the sequence 5′-AAG AGT AAT CTC CTG TCT CGA-3′ of RAB11A mRNA that correspond to nucleotides 165–185 of the Homo Sapiens mRNA transcript variants 1 and 2 deposited in NCBI (https://www.ncbi.nlm.nih.gov/nucleotide/NM_004663.5?report=genbank&log$=nuclalign&blast_rank=2&RID=Y7KDX6GN016 for transcript variant 1, and https://www.ncbi.nlm.nih.gov/nucleotide/NM_001206836.2?report=genbank&log$=nucltop&blast_rank=1&RID=Y7KDX6GN016 for transcript variant 2, accessed on 10 February 2023) with sequence ID NM_004663.5 and NM_001206836.2, respectively.

### 2.2. Cell Cultures Maintenance

The K562 cell line, a CML culture with BCR-ABL1 e14a2 fusion transcript, was obtained from the American Type Culture Collection (cell line reference: CCL-243; ATCC, Manassas, VA, USA), and maintained in Dulbecco’s modified eagle medium (DMEM, ThermoFisher Scientific, Waltham, MA, USA) supplemented with 10% (*v*/*v*) exosome depleted fetal bovine serum (FBS, ThermoFisher Scientific) and a mixture of 100 μg/mL Streptomycin and 100 U/mL Penicillin (ThermoFisher Scientific) at 37 °C, 5% (*v*/*v*) CO_2_ and 99% (*v*/*v*) relative humidity. For simplicity, from now on the supplemented medium will only be called exo-DMEM.

### 2.3. RAB11A Silencing in K562 Cell Line

For RAB11A silencing, 1 mL of K562 culture in a density of 1 × 10^5^ cells/mL was placed in the wells of a 24 wells plate and incubated with 0.45 nM AuNP@RAB11A (equivalent to 20 nM of oligonucleotide), 0.45 nM AuNP@PEG, or untreated (control) for 3, 6, 12, or 24 h. Afterwards, the culture was transferred to a clean tube and centrifuged for 500× *g* for 5 min, RT. The resultant supernatant was used for exosome purification and the pellet was used for evaluation of *RAB11A* mRNA expression by RT-qPCR or for Rab11a protein expression by Western-Blot. 

### 2.4. Cell Viability

The viability of cells after exposure to nanoformulations was accessed, as previously described [45]. In a first approach, 1 mL of 1 × 10^5^ K562 cells were distributed in wells from 24 well plates and exposed for 24 h to 0.45 nM AuNP@RAB11, AuNP@PEG or the same volume of Phosphate Buffer Saline (PBS). The quantification of viable cells was performed using Trypan blue (ThermoFisher Scientific), which is a dye that only enters cells with compromised membrane [46]. The percentage of viable cells after incubation with nanoformulations was calculated by comparison with cells treated with PBS. In parallel, cells were exposed to 1 μM Imatinib as positive control, or 0.1% (*v*/*v*) DMSO (vector control of imatinib).

In parallel, K562 cells were seeded in a 96-well plate with a density of 1 × 10^5^ cells/mL and exposed to the same stimulus, as described above, for the Trypan blue exclusion method. After 24 h, 20 μL of Cell Titer 96^®^ Aqueous one solution cell proliferation assay (Promega, Madison, WI, USA) was added, the absorbance at 490 nm was measured and Abs values were corrected to the respective solution without cells. Cell viability was calculated by normalizing to the respective control conditions, as explained above for the Trypan blue method.

### 2.5. Evaluation of RAB11A Expression in K562 Cell Line by RT-qPCR

The evaluation of *RAB11A* expression was performed by reverse transcriptase—quantitative polymerase chain reaction (RT-qPCR). The mRNA of the pelleted cells obtained after incubation with AuNPs was extracted with NZYol (NZYtech, Lisbon, Portugal) according to manufacturer’s instructions and cDNA was synthesized from 150 ng of total mRNA using NZY MuLV First-Strand cDNA Synthesis kit (NZYtech) with the protocol specified by the manufacturer. Expression of *RAB11A* was examined using the NZY qPCR green mastermix (NZYtech) and 0.5 μM primer forward (5′-AATCCCATCACCATCTTCCAG-3′) and 0.5 μM primer reverse (5′-GAGCCACACCATCCTAGTTG-3′) [27]. For RT-qPCR, a Corbett Rotor-Gene 6000 thermal cycler (Qiagen, Hilden, Germany) was used, with the following settings, 95 °C, 5 min, followed by 30 cycles of 95 for 45, 62 for 25 and 72 °C for 45 s. The *RAB11A* expression levels in AuNP@RAB11A treated cells was determined by the Ct method (2^−∆∆Ct^) [47] by normalization with housekeeping gene GAPDH [48] and with cells incubated with AuNP@PEG. 

### 2.6. Western-Blot for Evaluation of Rab11a Protein Expression in K562 Cell Line

The evaluation of Rab11a protein expression in K562 cells incubated for 12 h with 0.45 nM AuNP@RAB11A or AuNP@PEG by Western blot was performed according to protocols previously described [44] with the following modifications. Briefly, 10 μg total proteins were separated in a Sodium dodecyl sulfate polyacrylamide gel electrophoresis (SDS-PAGE) and transferred to a PVDF membrane (GE Healthcare, Chicago, IL, USA). After blotting, the membrane was stained with Ponceau S stain (Pierce, Appleton, WI, USA), followed by blocking with 5% (*w*/*v*) non-fat dairy milk in TBST (50 mM Tris-HCl, pH 7.5, 150 mM NaCl, 0.1% (*v*/*v*) Tween-20). The membrane was then incubated with 1:10,000 dilution of anti-Rab11a antibody [EPR7587(B)] (ab128913, Abcam, Cambridge, UK) and respective secondary antibody, striped with stripping buffer (0.1 M glycine, 20 mM magnesium acetate, 50 mM KCl, pH 2.2), blocked and incubated with anti-β-actin antibody (ref A5441, Sigma Aldrich, St Louis, MO, USA). The protein band intensity in each sample was calculated with FiJi software [49]. To calculate the percentage of Rab11a, the band intensity of each sample was normalized to the β-actin protein intensity in the same membrane and to the band intensity of AuNP@PEG treated cells.

### 2.7. Exosomes Isolation

The exosomes of the collected supernatants were isolated according to protocols previously described by our group [17]. Briefly, the supernatants were filtered with a 0.2 μm syringe filter and then mixed with Total Exosome isolation reagent (ThermoFisher Scientific) according to the manufacturer’s protocol. Pelleted exosomes were solubilized in 30 μL PBS and maintained at −80 °C. The total protein concentration in exosome suspensions was quantified using the Pierce 660 nm protein assay reagent (ThermoFisher Scientific) according to manufacturer’s instructions. For nanoparticle tracking analysis (NTA) of exosomes suspensions, the pellet after centrifugation with isolation reagent was suspended in 1 mL PBS, instead of 30 µL, and then analyzed in a Nanosight NS300 (Malvern Panalytical, Malvern, UK).

### 2.8. ELISA for Exosomes Characterization

For the enzyme-linked immunoassay (ELISA), the exosomes in samples incubated for 12 h with nanoformulations were analyzed with ExoELISA-ULTRA Complete Kit (CD63 detection) from Systems Biosciences (Palo Alto, CA, USA) according to manufacturer’s instructions, or by using a procedure adapted from Longatti et al. [50] (Alix and CD81 detection). Briefly, exosomes suspension in coating buffer, in a 20:100 proportion, were incubated o.n. in a MaxiSorp clear Flat-Bottom Immuno 96-Well Plates (ThermoFisher Scientific) to apply in each well the total amount of exosomes collected in the supernatant. After washing three times with PBST (PBS supplemented with 0.05% (*v*/*v*) Tween-20 (Sigma Aldrich)), the wells were incubated for 1h with blocking buffer (1% (*w*/*v*) bovine serum albumin (BSA, NZYtech) in PBST), followed by 2 h incubation with a dilution 1:500 of Alix Monoclonal Antibody (3A9, ref: MA1-83977, ThermoFisher Scientific) or Anti-CD81 antibody (M38, ref: ab79559, Abcam). The wells were washed three times for 5 min with PBST, incubated for 1 h with 1:1000 dilution of Anti-mouse IgG, HRP-linked Antibody (ref: 7076, Cell Signaling Technology, Danvers, MA, USA), washed three times for 5 min with PBST, incubated for 15 min with 1-step ultra TMB—ELISA (ThermoFisher Scientific) and the absorbance at 450 nm was measured after adding 2M sulfuric acid. For control purposes, it was also analyzed exo-DMEM, the supernatant collected before K562 exosomes purification, and K562 cell lysate at the same protein concentration as the one obtained for exosomes (0.5 µg/µL). The control samples were mixed with coating buffer in the 20:100 proportion used for exosomes suspensions analysis. To confirm the presence of the proteins in the exosome suspension, the Abs450 in samples was first corrected to the blank sample (coating buffer), followed by normalization to the results obtained for exo-DMEM sample. 

### 2.9. Ex-OVO Angiogenesis Assays

The experiments were made using procedures described before [17,41]. Briefly, after 72h incubation at 37 °C, 90% (*v*/*v*) relative humidity, the fertilized eggs (Pinto Valouro, Bombarral, Portugal) were opened to a weighing boat assuring that the embryo is facing upward. Black silicone O-rings (inside diameter 8 mm) were placed equidistantly above the blood vessels of the embryo and 40 μL of samples was added in O-rings assuring that each embryo was not subjected to the same set of samples, consisting in (1) PBS (control); (2) 1 × 10^9^ exosomes isolated from untreated K562; (3) 1 × 10^9^ exosomes isolated from K562 incubated for 12 h with 0.45 nM AuNP@RAB11A; (4) 1 × 10^9^ exosomes isolated from K562 incubated for 12 h with 0.45 nM AuNP@PEG; (5) a mixture of 1 × 10^9^ exosomes isolated from untreated K562 with 0.45 nM AuNP@RAB11A; (6) a mixture of 1 × 10^9^ exosomes isolated from untreated K562 with 0.45 nM AuNP@PEG; (7) 0.45 nM AuNP@RAB11A; and (8) 0.45 nM AuNP@PEG. The percentage of newly formed vessels was calculated, as previously described [41].

The chicken embryo mRNA analysis was performed according to procedures previously described in our group [17,42]. Briefly, each embryo was exposed to three O-rings containing the same stimulus, consisting of (1) 1 × 10^9^ exosomes isolated from untreated K562, (2) PBS (control stimulus 1), (3) 0.45 nM AuNP@RAB11A, (4) 0.45 nM AuNP@PEG (control stimulus 3), (5) 1 × 10^9^ exosomes isolated from K562 incubated for 12 h with 0.45 nM AuNP@RAB11A or (6) 1 × 10^9^ exosomes isolated from K562 incubated for 12 h with 0.45 nM AuNP@PEG (control stimulus 5). The expression of *IL8*, *VEGFA* or *FLT1* were analyzed using the 2^−∆∆Ct^ method, with *GAPDH* as internal control and the respective stimulus control, as above indicated.

### 2.10. Statistical Analysis

Results represent the average ± standard deviation of at least three independent experiments. Student’s *t*-test was performed using GraphPad prism vs. 7.0 (GraphPad software, Inc., San Diego, CA, USA). The difference between two values with *p*-value < 0.05 was considered statistically significant.

## 3. Results

The evaluation of the AuNP@RAB11 silencing efficiency was assessed on the CML cell line K562, before and after incubation with the AuNP@RAB11A nanoformulation via quantification of *RAB11A* mRNA expression, Rab11a protein expression, and by examining the exosome secretion. Afterwards, the effect of AuNP@RAB11A-exosomes (i.e., exosomes retrieved from K562 leukemic cells after silencing with the nanoformulation) on the in vivo neoangiogenesis was assessed via the analysis of formation of new vessels in a CAM model.

### 3.1. RAB11A mRNA Silencing with AuNP@RAB11A

For *RAB11A* silencing, an shDNA targeting the anti-RAB11A mRNA was designed using a specific recognition sequence previously proposed by Lipinski and coworkers [43] as guidance. This shDNA was then used to functionalize AuNPs previously covered with 30% PEG (AuNP@PEG), forming the AuNP@RAB11A [51,52]. UV-Vis spectroscopy and dynamic light scattering (DLS) were performed to characterize functionalization of AuNPs (Appendix A), as previously described [51,52]. The silencing efficacy of the nanoconjugates was then determined by challenging K562 cells with 0.45 nM of the nanoconjugates (corresponding to 20 nM of oligo) for 3, 6, 12 and 24 h. Total RNA was extracted and analyzed, as described elsewhere [44]. The relative expression of *RAB11A* in AuNP@RAB11A-treated samples was compared with the expression of samples treated with the corresponding gold concentration of AuNP@PEG (0.45 nM) using *GAPDH* gene expression as internal control (Figure 1a). Results show that incubation of K562 cells with AuNP@RAB11A resulted in a 40% decreased expression of *RAB11A* after 6 h, that was still observable up to 24 h (Figure 1a). The decreased gene expression is also reflected in protein expression (Figure 1). Western blot analysis revealed that relative expression of Rab11a protein was significatively reduced 14.5 ± 3.5% when K562 cells were incubated for 12 h with AuNP@RAB11A compared to AuNP@PEG-treated cells (Figure 1b,c, Appendix A).

To understand if the nanoformulations have some effect in cell viability, the cell proliferation was evaluated using the 3-(4,5-dimethylthiazol-2-yl)-5-(3-carboxymethoxyphenyl)-2-(4-sulfophenyl)-2H te-trazolium, inner salt (MTS) colorimetric assay or the Trypan blue exclusion method. No alterations were observed when cells are incubated with AuNP@RAB11 or AuNP@PEG relative to untreated cells (Appendix A), suggesting that nanoformulations or the silencing has no effect on cell proliferation or viability.

### 3.2. Exosomes Secreted by K562 Treated with AuNP@RAB11A Are Smaller and Present Different Protein Content than K562 Exosomes Counterparts

First, we evaluated the pattern of exosome secretion through time upon incubation with anti-*RAB11A* nanoconjugate. Cells were incubated for 3, 6, 12 or 24 h with AuNP@PEG or AuNP@RAB11A and then supernatant was collected via centrifugation to remove cells in suspension. After exosome collection, the protein content in each exosomal fraction was measured and the obtained amount in AuNP@RAB11A treated samples was compared to the amount of AuNP@PEG (Figure 2A) [18]. Results suggest that during the first 12 h, cells incubated with AuNP@RAB11A gradually secrete less exosomes than AuNP@PEG treated cells, and after 24 h the amount of protein is similar in exosomes suspensions collected from cells exposed to both AuNPs (Figure 2A). A decreased protein concentration was detected after 12 h incubation (Figure 2A), suggesting that silencing with AuNP@RAB11A results in a decreased exosome secretion in this time period. Hence, further analysis was only focused on exosomes secreted after 12 h incubation with nanoformulations. The size and exosome concentration were then inferred by nanotracking analysis (Figure 2B). Interestingly, a higher concentration of exosomes in AuNP@PEG and AuNP@RAB11A-treated cells relative to the untreated cells (Figure 2B) was observed, which might be associated with stimulation of the endocytic pathway due to AuNP internalization [53,54]. Despite no significant alterations observed in the concentration of AuNP@RAB11A treated exosomes relative to AuNP@PEG treatment, a slight decrease in the size of exosomes secreted after incubation with nanoformulations containing the anti-*RAB11A* (Figure 2B) was observed. Although it would be tempting to hypothesize that the deviation of the peak occurred due to the presence of 30.5 ± 0.2 nm (Appendix A) AuNPs (Appendix A), a decrease in the size of AuNP@PEG-treated K562 exosomes (Figure 2B) was not observed, and the NTA peak of AuNP@RAB11A treated K562 exosomes is centered at 90 ± 16 nm as can be observed in the representative NTA spectra showed in Figure 2C. The difference between exosomes of different conditions, was also observed when the presence of the exosome markers CD63, CD81 and Alix were inferred by ELISA, being observed an increased intensity of Alix in AuNP@RAB11A-treated K562 exosomes relative to the other two conditions (Figure 2D and Appendix A). Overall, the evaluation of exosomes secretion suggests that silencing of RAB11A mRNA in K562 cells, results in the release of smaller exosomes (Figure 2B) with increased amounts of Alix protein (Figure 2D).

### 3.3. The Pro-Angiogenic Potential of Exosomes Secreted by K562 Treated with AuNP@RAB11A Is Lower than the Pro-Angiogenic Potential of K562 Exosomes Counterparts

The evaluation of the effect of exosomes secreted by AuNP@RAB11A incubated cells (silenced exosomes) on angiogenesis was performed on the CAM model as previously described by our group [17,41]. O-rings were equidistantly placed and filled with PBS for control purposes, AuNP@RAB11A, AuNP@PEG, K562 exosomes collected after 12 h incubation with fresh medium (untreated K562 exosomes), a mixture of K562 exosomes and AuNP@RAB11A or AuNP@PEG, or with K562 exosomes collected after 12 h incubation with AuNP@RAB11A (silenced) or with AuNP@PEG. The number of vessels that sprout from higher caliber veins was counted at 0 and 24 h timepoints, and the number of newly formed vessels was calculated after normalization to the number of lower caliber vessels in PBS treated CAM regions (Figure 3a,b, and Appendix A). In agreement with results previously described [15,17], K562 derived exosomes induced the formation of tertiary vessels (Figure 3c and Appendix A). The incubation of the CAM with AuNP@RAB11A did not affect the number of newly formed vessels, i.e., when CAMs were only exposed to NPs, the number of tertiary vessels was identical to the control CAMs, while the simultaneous exposure of the CAMs to K562 exosomes and AuNP@RAB11A retrieved a similar increase of tertiary vessels to the K562 exosomes incubated CAMs (Figure 3). Interestingly, the CAM incubation with silenced exosomes did not trigger angiogenesis (Figure 3c and Appendix A). 

Due to the important role of Rab11 in the transport of VEGR-1 and VEGFR-2 from recycling vesicles to the plasma membrane [20,21,23], we further explored if the effect of AuNP@RAB11A treated exosomes of neoangiogenesis in CAMs could be correlated with the VEGFR pathway. Previous studies performed by our group revealed that K562 exosomes induced neo-angiogenesis within the first hours of incubation via *VEGFR1* dependent pathway [17]. In fact, 12 h exposure to K562 exosomes resulted in a 200-fold increase of *VEGFR1*, with no significant alteration of *VEGFA* or *IL8* [17]. Hence, the expression of these genes was evaluated after 12 h exposure of the CAM to K562 exosomes, PBS (control of K562 exosomes), AuNP@RAB11A, AuNP@PEG (control of AuNP@RAB11A), or with K562 exosomes collected after 12 h incubation with AuNP@RAB11A or with AuNP@PEG (control of AuNP@RAB11A incubated exosomes). Interestingly, no *VEGFR1* mRNA amplification was detected, while an increased expression of *IL8* and *VEGFA* when CAMs were exposed to silenced K562 exosomes (Figure 4). The same trend was observed in AuNP@RAB11A-treated CAMs, where a decreased expression of *VEGFR1* and increased expression of *VEGFA* was observed (Figure 4). These are interesting results, since no major alterations in the number of newly formed vessels were detected after CAMs incubation with AuNP@RAB11A (Figure 3), suggesting that silenced exosomes induce similar effect than nanoformulations alone, but with higher impact.

## 4. Discussion

When in the active GTP-bound conformation, Rab small G proteins recruit effector proteins that act in cargo selection, vesicle formation from donor membranes, vesicle transport through cytoskeleton, and vesicle fusion with receptor membranes or transfer to another Rab protein [55]. The Rab11 family, composed of Rab11a, Rab11b and Rab25 proteins, are major regulators of the exocytic and recycling processes, by regulating protein and vesicle formation and transport from early and recycling endosomes to the cell surface [18,55]. Despite sharing high sequence homology, Rab11b protein is mainly expressed in brain, heart and testis, Rab25 expression is restricted to epithelial cells, and Rab11a is expressed ubiquitously [55,56]. Previous studies showed that depletion of Rab11a is non-lethal [57], but have an important role in the angiogenic process, via vascular endothelial-cadherin recycling [22,57], recycling of the NRP-1 protein [58], recycling of VEGFR-2 via NRP-1 [20,23], and recycling of α5-integrin-p-FAK to promote assembling of adhesion sites [25]. Moreover, an overexpression of the Rab11 family-interacting protein 2 (Rab11-FIP2) in colorectal cancer patients was correlated with increased angiogenesis, tumor migration and consequent metastasis formation [59]. The role of Rab11 in the regulation of the exosome pathway also increment to the importance of these proteins for tumor progression, angiogenesis and metastases [26,60]. Particularly, Rab11 is involved in the formation and secretion of exosomes in CML cell lines [27], that exhibit a pro-angiogenic effect in human vascular endothelial cells (HUVEC) [15,16] and in CAM of the chicken embryo [17]. Herein, we aimed at evaluating *Rab11A* as a target for gene silencing to curb neoangiogenesis mediated by leukemic exosomes in vivo. In fact, we successfully downregulated Rab11A via AuNPs functionalized with an anti-*RAB11A* hairpin in the leukemia K562 cells, attaining a staggering 40% decrease of mRNA expression after 6 h incubation, which endure up to 24 h (Figure 1a). Consequently, Rab11a protein expression decreased 14 ± 3.5% after 12 h (Figure 1b), which led to the secretion of exosomes with smaller size (Figure 2B) and different protein content (Figure 2D).

We previously showed the effect of CML-derived exosomes on angiogenesis using a CAM model, where it was observed a time and concentration dependent effect on the increased number of newly formed low caliber vessels [17]. Herein, results corroborate those findings where an increased number of newly formed vessels is observed after 24 h exposure to K562 exosomes (Figure 3). The cell-to-cell communication highway provided by exosomes seems to be crucial for the effective impact from the molecular drivers of neoangiogenesis. In fact, when CAMs are incubated with silenced exosomes (secreted by K562 cells exposed for 12 h to AuNP@RAB11A), the number of newly formed vessels was comparable to that of PBS (Figure 3). This suggests that the pro-angiogenic potential of K562-exosomes is counteracted by *RAB11A* silencing. Interestingly, the incubation of the CAM with AuNP@RAB11A did not retrieve major alterations, suggesting that nanoformulations per se had no effect on the formation of new vessels. This is rather interesting, considering that the selected anti-*RAB11A* sequence shows 100% identity with *Gallus gallus RAB11A* mRNA [61] and, if the AuNP@RAB11A had been taken up by the embryo, a certain degree of gene silencing with consequent phenotypic alteration would be expected. Since it was described a direct correlation between *VEGFR* mRNA expression, protein abundance and vein phenotype [62,63], it was explored the expression of angiogenesis-related genes *IL8*, *VEGFA* or *VEGFR1* after 12 h incubation of CAMs with AuNP@RAB11, which revealed similar expression alterations, although in a lower extent, than the ones observed in CAMs incubated with silenced exosomes (Figure 4). In fact, no *VEGFR1* mRNA expression was detected in CAMs incubated with silenced exosomes, and a 0.1 ± 0.02-fold decrease was observed after exposure to AuNP@RAB11 (Figure 4). These results are in line with the described role of Rab11 in the transport of VEGFR [20,22,23,57], suggesting that silenced *RAB11A* will result in lowered expression of *VEGFR1.* It is feasible that this decrease will be responsible for the observed increase expression of *VEGFA* and *IL8*, observed in CAMs incubated with silenced exosomes, possibly to sustain neo-angiogenesis. In fact, although, to our knowledge, it was never described a correlation between *VEGFR1* decreased expression and consequent *IL8* increased expression, the expression of both genes seems to be tightly regulated, with several studies reporting a correlation between VEGF- and IL8-mRNA expression in human breast cancer [64], human non-small-cell lung cancer [65], human malignant astrocytomas [66], or in human head and neck squamous carcinoma cell lines [67].

Our data suggests that the decreased expression of Rab11a in tumor cells will alter their composition, which might also result in the secretion of exosomes with lower pro-tumoral effect. This statement is supported by the fact that alteration of the expression of Rab27a and Rab27b proteins, involved in the transport of late endosomal/lysosomal-like compartments to the plasma membrane in the exosomal pathway [26], result in an alteration of the tumor-cell-derived exosomes tumoral effect [68,69,70]. Overexpression of Rab27a in the non-small-cell lung cancer cell line A549 prompted antitumor immunity, by upregulating the major histocompatibility complex II molecules and promoting the expression of antitumor type I cytokines [69]. Another study showed the pro-tumoral effect of exosomes secreted by breast cancer cell line MCF-7 in normal human bronchial-tracheal epithelial cells, observing the increased expression of *C-MYC* oncogene after exosomes internalization [70]. The *C-MYC* overexpression in bronchial-tracheal epithelial cells was not observed after internalization of MCF-7 derived exosomes incubated with AuNPs functionalized with an anti-RAB27A hairpin [70]. 

All in all, silencing of Rab proteins involved in exosome biogenesis yield exosomes with lower pro-tumoral effect in vivo, which may have big implications on how we tackle cancer development. In fact, herein we demonstrate the potential of nanomedicine to specifically target molecular pathways involved in exosome biogenesis and intracellular transport, which may open avenues towards an anti-tumor therapy based on the decreased tumoral effect of exosomes, be it in downstream malignant transformation of naïve cells or by modulating cancer progression in invasion and metastasis. 

## Figures and Tables

**Figure 1 pharmaceutics-15-00742-f001:**
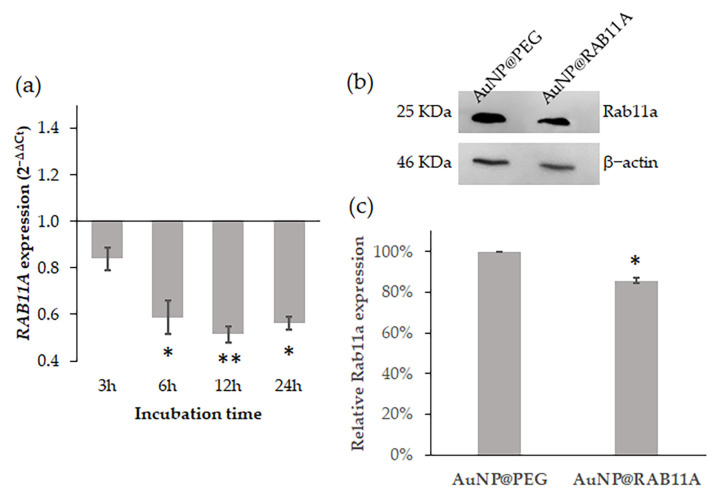
Evaluation of silencing efficacy of AuNP@RAB11A. (**a**) Time course of *RAB11A* gene expression in K562 cells incubated with 0.45 nM AuNP@RAB11A. Gene expression variation was calculated with 2^−∆∆Ct^, using as reference *GAPDH* gene and AuNP@PEG treated sample; (**b**) Western blot of Rab11a protein and β-actin protein in the same membrane, of K562 cells incubated for 12 h with 0.45 nM AuNP@PEG or AuNP@RAB11A; (**c**) Percentage of Rab11a protein relative intensity values normalized to β-actin protein intensity in the same membrane and to the AuNP@PEG control sample. Bars represent the average and error bars represent the standard deviation of four independent experiments. * *p*-value < 0.05; ** *p*-value < 0.005 relative to respective AuNP@PEG.

**Figure 2 pharmaceutics-15-00742-f002:**
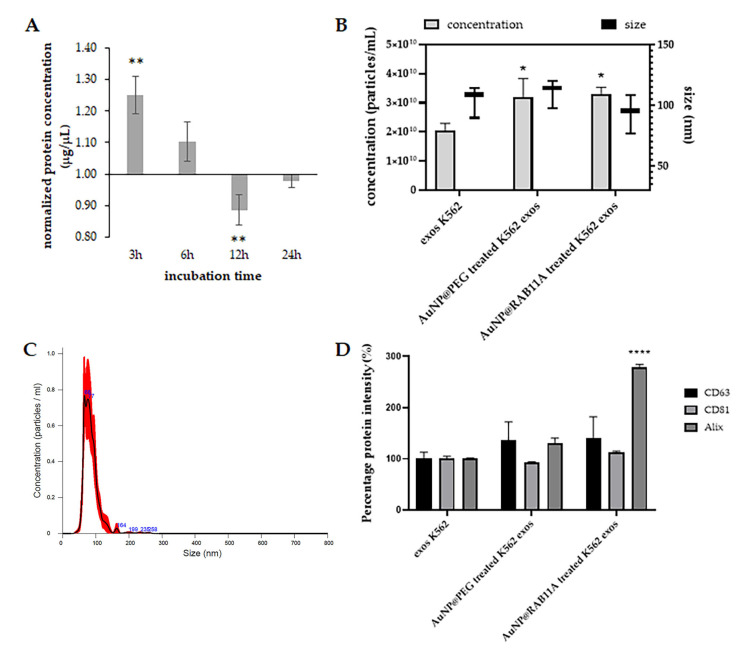
Evaluation of exosome secretion after incubation with AuNP@RAB11A. (**A**) Time course of protein concentration in exosome suspensions secreted by K562 cells incubated with 0.45 nM AuNP@RAB11A relative to cells incubated with AuNP@PEG. Bars represent the average and error bars the standard deviation of at four independent experiments; (**B**) Nanotracking analysis (NTA) to infer exosome concentration (left y axis, results are represented by grey bars ± SEM of three independent experiments) and size (right y axis, results are represented by whiskers plots of three independent experiments) of K562 cells incubated for 12 h with 0.45 nM AuNP@PEG or AuNP@RAB11A, or untreated (exos K562); (**C**) representative NTA spectra obtained for exosomes collected from K562 cells, after 12 h incubation with 0.45 nM AuNP@RAB11A; (**D**) Percentage of protein intensity of CD63 (black bars), CD81 (light grey bars) or ALIX (dark grey bars) in exosomes of K562 cells incubated for 12 h with 0.45 nM AuNP@PEG or AuNP@RAB11A, or untreated (exos K562). Values were obtained by ELISA and normalized to the values obtained for exos K562. Bars represent the average and error bars the standard deviation of three independent experiments. * *p*-value < 0.05 relative to control; ** *p*-value < 0.005 relative to AuNP@PEG-treated samples; **** *p*-value < 0.0001 relative to control.

**Figure 3 pharmaceutics-15-00742-f003:**
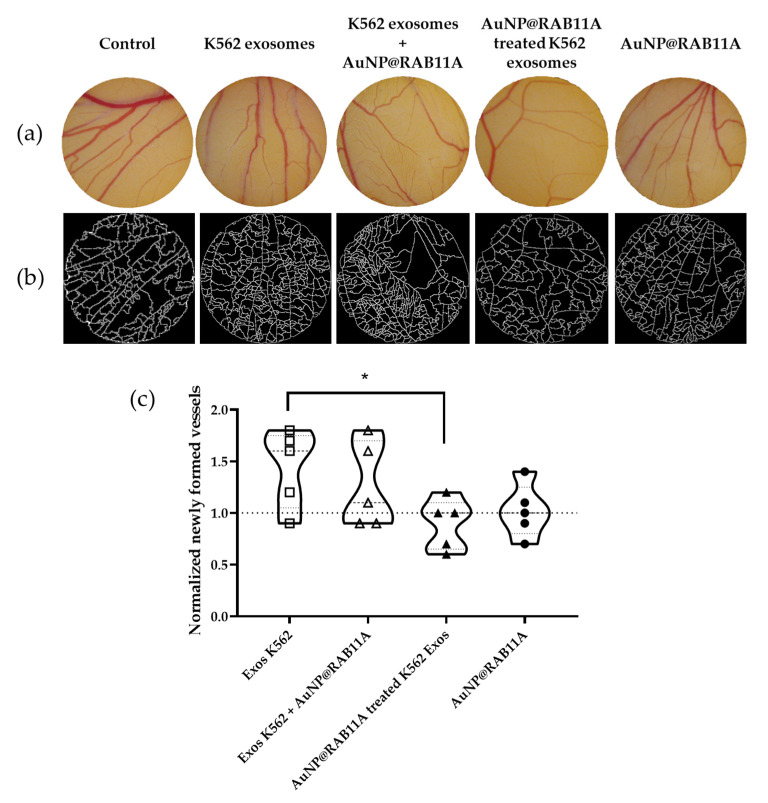
Evaluation of angiogenic potential of K562 exosomes (1 × 10^9^ exosomes; Exos K562), a mixture of K562 exosomes (1 × 10^9^ exosomes) with 0.45 nM AuNP@RAB11A (Exos K562 + AuNP@RAB11A), 1 × 10^9^ exosomes collected from K562 exposed for 12 h to 0.45 nM AuNP@RAB11A (AuNP@RAB11A treated K562 exos), or to 0.45nM AuNP@RAB11A. (**a**) Aspect of the chorioallantoic membrane (CAM) exposed for 24 h with control (Phosphate buffer saline, PBS), Exos K562, Exos K562 + AuNP@RAB11A, AuNP@RAB11A treated K562 exos or AuNP@RAB11A with 4× magnification. (**b**) Segmented image of the respective CAM region used to calculate number of branches; (**c**) Violin density plots of five independent experiments of newly formed vessels in CAMs exposed for 24 h to Exos K562 (each independent experiment represented as squares in the graph), Exos K562 + AuNP@RAB11A (each independent experiment represented as white triangles in the graph), AuNP@RAB11A-treated K562 exos (each independent experiment represented as dark triangles in the graph) or AuNP@RAB11A (each independent experiment represented as dark circles in the graph), normalized to CAM regions incubated with vector control (PBS) and to the respective CAM at 0 h timepoint. Dotted line at 1.0 normalized newly formed vessels refers to control sample (region of the CAM incubated with PBS after 24 h normalized to respective CAM at 0 h). * *p*-value < 0.05.

**Figure 4 pharmaceutics-15-00742-f004:**
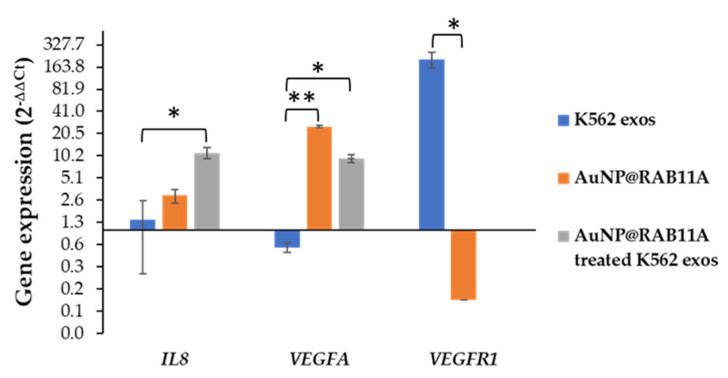
*IL8*, *VEGFA* and *VEGFR1* mRNA expression levels after 12 h of chorioallantoic membrane (CAM) exposure to 1 × 10^9^ exosomes from K562 cells (K562 exos, blue bars), 0.45 nM AuNP@RAB11A (orange bars), or 1 × 10^9^ exosomes from K562 cells incubated for 12 h with AuNP@RAB11A (AuNP@RAB11A treated K562 exos, grey bars). Data were normalized to the GAPDH mRNA levels, followed by normalization to PBS treated CAMs, 0.45 nM AuNP@PEG treated CAMs, or to 1 × 10^9^ exosomes from K562 cells incubated for 12 h with AuNP@PEG, respectively. * *p*-value < 0.05; ** *p*-value < 0.005 relative to respective mRNA expression in K562 exos sample.

## Data Availability

The data can be shared up on request.

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
