# Peer review of "Exploring *RAB11A* Pathway to Hinder Chronic Myeloid Leukemia-Induced Angiogenesis In Vivo"

_pharmaceutics, 2023, doi:10.3390/pharmaceutics15030742_

Round 1

Reviewer 1 Report

In the paper entitled “Exploring RAB11A Pathway to Hinder Chronic Myeloid Leukemia Induced Angiogenesis in vivo” the authors reported the use of gold nanoparticles functionalizated with anti-RAB11A oligonucleotides to downregulate RAB11A mRNA in a leukemia cell line, observing high silencing values of mRNA and associated proteins. The angiogenic potential of the exosomes secreted by the functionalizated nanoparticles incubated cells was also analyzed. In general, I found the paper suitable for its publication in Pharmaceutics journal after minor revisions. In my opinion, the introduction is very well-organized and detailed, clearly presenting the state of the start in the field and justifying the novelty of the performed work. Resultas and discussion sections are also well-constructed, with the adequate references to support the reached conclusions. Following I expose some minor comments and suggestions that could improve the paper and which I would like the authors to address before consider resubmission.

- I suggest the ellaboration of a simple schematic figure or graphical abstratct to illustrate the performed work.

- Avoid the definition of abbreviations only used once during the abstract (VEGFR).

- The only thing that I missed in the introduction section is a better motivation about the use of gold nanoparticles. Why gold and not other organic/inorganic supports to perform the functionalization with oligonucleotides? Is it for their photothermal capability? Ellaborate also a little bit the origin and potential application of this photothermal properties inherennt to gold nanoparticles. You can see for instance https://doi.org/10.1186/s12951-019-0538-3, https://doi.org/10.1021/jp505979e or  https://doi.org/10.1021/acs.molpharmaceut.9b00021

- Line 248: use the same significance in main value and standard deviation, that is, 14.0 ± 3.5.

- Fig. 2A: please explain why decreased the relative amount of secreted exosomes from 12 h to 24 h.

Reviewer 2 Report

Angiogenesis may result from chronic myeloid leukemia(CML). The author used gold nanoparticles (AuNPs) with an anti-RAB11A oligonucleotide (AuNP@RAB11A) to downregulate RAB11A mRNA level in K562 CML cells, which showed a 40% silencing of the mRNA. In CAM model, these exosomes secreted by AuNP@RAB11A incubated K562 cells did not present the angiogenic potential of those secreted from untreated K562 cells. These data herein suggested that the relevance of Rab11 for the angiogenesis mediated by leukemia exosomes, whose deleterious effect may be overcome by silencing of RAB11. Basically, this manuscript is interesting and suggestive. There some concerns should be addressed before publication: Major points: 1.The collected data used classical CAM model to observe angiogenesis. They did not perform that in more higher species, such as in mice or rats, which is much closer to human. Moreover, BCR-ABL transgenic mouse was usually administrated in CML study. 2.If rodent animal experiments can not be administrated, perhaps they should try use cells from CML patients to prepare exosome to verify their findings in angiogenesis. Minor points: 1. Author should provide the control sequence of Anti-sense hairpin. 2. Please add the ATCC number of K562 cell lines. 3.In Fig.1, the sample size was not provided in figure legend. 4. In Fig.2, the sample number was missed in figure legend. 5.In Fig.3, there are no sample numbers in figure legend. 6.In Fig.4, ELISA should be conducted in addition to the q-PCR analysis.

Author Response

Angiogenesis may result from chronic myeloid leukemia(CML). The author used gold nanoparticles (AuNPs) with an anti-RAB11A oligonucleotide (AuNP@RAB11A) to downregulate RAB11A mRNA level in K562 CML cells, which showed a 40% silencing of the mRNA. In CAM model, these exosomes secreted by AuNP@RAB11A incubated K562 cells did not present the angiogenic potential of those secreted from untreated K562 cells. These data herein suggested that the relevance of Rab11 for the angiogenesis mediated by leukemia exosomes, whose deleterious effect may be overcome by silencing of RAB11. Basically, this manuscript is interesting and suggestive. There some concerns should be addressed before publication: Major points:

Q1. The collected data used classical CAM model to observe angiogenesis. They did not perform that in more higher species, such as in mice or rats, which is much closer to human. Moreover, BCR-ABL transgenic mouse was usually administrated in CML study.

A1.  We thank reviewer’s valuable comment. In fact, the use of rodents, particularly a BCR-ABL transgenic mouse would give important results that will certainly improve this study, especially to understand the effect of RAB11A silenced exosomes in the CML bone marrow maturation.  However, the major importance of this manuscript is that describes, using an in vivo model, that the angiogenic potential of RAB11A silenced exosomes is lower than the pro-angiogenic effect of K562 secreted exosomes. We consider that CAM assay is suitable for the purpose of the study, since it is a functional assay that allow to easily observe an effect of exosomes in neoangiogenesis. Moreover, CAM assay is a well-recognized and simple model for angiogenesis, as revised in several papers (e.g. 10.1089/ten.teb.2020.0048, 10.3390/ijms23010452, or 10.1021/acsbiomaterials.9b00172), that obeys to the 3Rs policy for the use of animal models.

Q2. If rodent animal experiments can not be administrated, perhaps they should try use cells from CML patients to prepare exosome to verify their findings in angiogenesis.

A2. We understand reviewer’s point of view. In fact, the progression of CML is highly correlated to genomic instability, resulting in a heterogeneous group of cells with different genomic traits as described in Abdulmawjood et al, 2021 (doi: 10.3390/ijms222212516). Since exosomes content is cell dependent (Hamzah et a, 2021, doi: 10.3390/ijms22105346), it is expected that cells from different patients will secrete exosomes with different content and different angiogenic potential. Nevertheless, as referred in lines 41-44 of the manuscript, the group of Prof Ricardo Alessandro used CML cells and blood samples of CML patients to evaluate the effect of exosomes in the maturation of the bone marrow and found that they are important players in neoangiogenesis promotion (doi:10.1111/jcmm.12873, doi:10.1002/ijc.26217). We are aware that, due to the inherent complexity of the organism, this study only unveils a little tip of the iceberg on the role of Rab11A in the angiogenic effect of K562 exosomes. Nevertheless, the submitted study is a proof of concept that aims the analysis of the effect of silencing RAB11A in exosomes secretion, and their further role in tumor progression. Hence, in a first approach, it is required the application of a well-known CML cell model such as K562, to evaluate the silencing efficacy of AuNP@RAB11A and following effect of K562 secreted exosomes, which are well known to promote angiogenesis as previously described in doi:10.1007/s10456-011-9241-1 or doi:10.2147/IJN.S215711.

Minor points:

Q3. Author should provide the control sequence of Anti-sense hairpin.

A3. The information about the target sequence of RAB11A mRNA was included in the manuscript in lines 105-113: “The antisense hairpin was designed to hybridize with the sequence 5’-AAG AGT AAT CTC CTG TCT CGA-3’ of  RAB11A mRNA that correspond to nucleotides 165-185 of the Homo Sapiens mRNA transcript variants 1 and 2 deposited in NCBI (https://www.ncbi.nlm.nih.gov/nucleotide/NM_004663.5?report=genbank&log$=nuclalign&blast_rank=2&RID=Y7KDX6GN016 for transcript variant 1, and  https://www.ncbi.nlm.nih.gov/nucleotide/NM_001206836.2?report=genbank&log$=nucltop&blast_rank=1&RID=Y7KDX6GN016 for transcript variant 2, accessed on 10th February 2023) with sequence ID NM_004663.5 and NM_001206836.2, respectively.”

Q4. Please add the ATCC number of K562 cell lines.

A4. The ATCC number of K562 cell lines were included in the text (line 116): “(cell line reference: CCL-243; ATCC…)”

 Q5. In Fig.1, the sample size was not provided in figure legend.

A5. The exact number of experiments were introduced in the figure legend, line 278: “Bars represent the average and error bars the standard deviation of four independent experiments.” The original images of 4 independent western-blots were submitted with the new manuscript as requested by the editor.

Q6. In Fig.2, the sample number was missed in figure legend.

A6. The exact number of experiments were introduced in the figure legend. Figure 2A – in line 325-326: “Bars represent the average and error bars the standard deviation of four independent experiments”; Figure 2B – the exact number was already submitted in the first version of the manuscript: Lines 327-329: “(left y axis, results are represented by grey bars ± SEM of three independent experiments) (…) (right y axis, results are represented by whiskers plots of three independent experiments); Figure 2c – it is a representative graphic of the three NTA experiments performed and represented in Figure 2b; Figure 2D – Lines 334-335: “Bars represent the average and error bars the standard deviation of three independent experiments.”

Q7.In Fig.3, there are no sample numbers in figure legend.

A7. The sample number was already informed in the figure legend in the first version of the manuscript. More, the violin plots were presented with the individual values of each experiment. To better explain this, the legend was modified (Lines 369-373) to: “Violin density plots of five independent experiments of newly formed vessels in CAMs exposed for 24h to Exos K562 (each independent experiment represented as squares in the graph), Exos K562 + AuNP@RAB11A (each independent experiment represented as white triangles in the graph), AuNP@RAB11A treated K562 exos (each independent experiment represented as dark triangles in the graph) or AuNP@RAB11A (each independent experiment represented as dark circles in the graph)…”

Q8. In Fig.4, ELISA should be conducted in addition to the q-PCR analysis.

A8. Another Reviewer had a related question, and we did our best to accommodate both views. We agree that values of mRNA and respective protein levels may not match. However, we have found that the available VEGFR antibodies in chicken are highly unspecific, with the major commercial antibodies aiming at human and rodent species. In fact, the VEGFR antibody that reacts with chicken (https://www.novusbio.com/products/vegfr1-flt-1-antibody_nb100-527), presents no specificity towards VEGFR1, since it also detects VEGFR2, and only 94% activity towards this species.

Nevertheless, a direct correlation between VEGFR and VEGF transcript, protein levels and vein phenotypic features has been reported (e.g. doi:10.1038/cdd.2009.152 and doi: 10.1038/s41598-019-43185-8). Hence, since we analysed the phenotype, resultant from protein abundance, after incubation with exosomes and nanoparticles, we may extrapolate with relative confidence that the transcription levels of genes coding proteins involved in the angiogenesis mechanism might have a direct correlation with the protein expression and resultant phenotype. This was clarified in the text in Line 449-450: “Since it was described a direct correlation between VEGFR mRNA expression, protein abundance and vein phenotype [61,62], it was explored the expression…”

Reviewer 3 Report

Review comments:

This study by Catarina Roma-Rodrigues et al presents an antisense oligonucleotide-functionalized gold nanoparticle (AuNP@RAB11A) to downregulate small GTP-binding protein Rab11a mRNA expression in the CML cell line K562. In an in vivo CAM model, the angiogenic potential of exosomes secreted by AuNP@RAB11A-treated K562 cells is diminished, and the molecular mechanism may be related to the downregulation of the VEGFR pathway. This study demonstrates the potential of developing nanomedicine targeting the synthesis and transport of exosomes for antitumor therapeutic effects.

However, there are several problems and suggestions to given.

Major concern:

1) In the manuscript, all significant differences are indicated using * p-value < 0.05. If a more precise representation (e.g. ** p-value < 0.01, *** p-value < 0.001) could be used, the significant differences between the results would be clearer.

2) The subtitle of the results section should reflect the results of the experiment, rather than simply summarize the experiment.

3) In the text corresponding to Figure 2, some concluding statements could be added to explain more clearly what the experimental results are trying to illustrate.

4) In Figure 3, a CAM model without any treatment should be added as a blank control to demonstrate the angiogenesis without treatment.

5) In Figure 4, the downregulatory effect of AuNP@RAB11A on the VEGFR pathway was verified only by the expression level of mRNA. The results would be more reliable if further validated at the protein level.

Minor concern:

1) In the supplementary material, Figure S5 is mislabeled as Figure S3, and this should be corrected in the corresponding text.

2) In the manuscript, there are some expressions that need to be harmonized, such as the writing of CD63, CD81, Alix of Figure 2D, Figure S4 and its figure legend; the first letter of ‘k562’ in line 368 should be capitalized; words of Latin origin such as in vivo in line 449 should be italicized.

3) In Figure S5, the ‘AuNP@PEG treated K562 exos’ in figure legend is mislabeled as ‘Exos c/ AuNP@PEG’ in the figure.

Author Response

This study by Catarina Roma-Rodrigues et al presents an antisense oligonucleotide-functionalized gold nanoparticle (AuNP@RAB11A) to downregulate small GTP-binding protein Rab11a mRNA expression in the CML cell line K562. In an in vivo CAM model, the angiogenic potential of exosomes secreted by AuNP@RAB11A-treated K562 cells is diminished, and the molecular mechanism may be related to the downregulation of the VEGFR pathway. This study demonstrates the potential of developing nanomedicine targeting the synthesis and transport of exosomes for antitumor therapeutic effects.

However, there are several problems and suggestions to given.

Major concern:

Q1) In the manuscript, all significant differences are indicated using * p-value < 0.05. If a more precise representation (e.g. ** p-value < 0.01, *** p-value < 0.001) could be used, the significant differences between the results would be clearer.

A1) The significance in each figure was adjusted as requested and significance meaning was included in the respective figure caption. The statistical significance of exosomes concentration in Figure 2B was also included since it was not presented in the first version of the manuscript by accident.

Q2) The subtitle of the results section should reflect the results of the experiment, rather than simply summarize the experiment.

A2) The subtitles of the results section were altered as suggested by reviewer:

3.1. RAB11A mRNA silencing with AuNP@RAB11A

3.2. Exosomes secreted by K562 treated with AuNP@RAB11A are smaller and present different protein content than K562 exosomes counterparts

3.3. The pro-angiogenic potential of exosomes secreted by K562 treated with AuNP@RAB11A is lower than the pro-angiogenic potential of K562 exosomes counterparts

Q3) In the text corresponding to Figure 2, some concluding statements could be added to explain more clearly what the experimental results are trying to illustrate.

A3) A further explanation and concluding statements were included in the text as requested by the reviewer (and to accommodate a related query by another reviewer). In lines 295-298, results obtained in Figure 2A were explained: “Results suggest that during the first 12h, cells incubated with AuNP@RAB11A gradually secrete less exosomes than AuNP@PEG treated cells, and after 24h the amount of protein is similar in exosomes suspensions collected from cells exposed to both AuNPs (Figure 2A).” In lines 317-320 the following phrase was also included: “Overall, the evaluation of exosomes secretion suggests that silencing of RAB11A mRNA in K562 cells, results in the release of smaller exosomes (Figure 2B) with in-creased amount of ALIX protein (Figure 2D).”

Q4) In Figure 3, a CAM model without any treatment should be added as a blank control to demonstrate the angiogenesis without treatment.

A4) The CAM of the control sample (incubated with PBS) was also included in Figure 3.

Q5) In Figure 4, the downregulatory effect of AuNP@RAB11A on the VEGFR pathway was verified only by the expression level of mRNA. The results would be more reliable if further validated at the protein level.

A5) We tried to accommodate the information also in response to a related query by another Reviewer. We agree that values of mRNA and respective protein levels may not match. VEGFR antibodies in chicken are highly unspecific, where the major commercial antibodies aim at human and rodent species. The VEGFR antibody that reacts with chicken (https://www.novusbio.com/products/vegfr1-flt-1-antibody_nb100-527), presents no specificity towards VEGFR1, since it also detects VEGFR2, and only 94% activity towards this species.

Nevertheless, a direct correlation between VEGFR and VEGF transcript, protein levels and vein phenotypic features has been reported (e.g. doi:10.1038/cdd.2009.152 and doi: 10.1038/s41598-019-43185-8). Hence, since we analysed the phenotype, resultant from protein abundance, after incubation with exosomes and nanoparticles, we may extrapolate with relative confidence that the transcription levels of genes coding proteins involved in the angiogenesis mechanism might have a direct correlation with the protein expression and resultant phenotype. This was clarified in the text in Line 449-450: “Since it was described a direct correlation between VEGFR mRNA expression, protein abundance and vein phenotype [61,62], it was explored the expression…”

Minor concern:

Q6) In the supplementary material, Figure S5 is mislabeled as Figure S3, and this should be corrected in the corresponding text.

A6) The figure number was corrected in Supplementary materials and in the text (Lines 350, 352 and 358).

Q7) In the manuscript, there are some expressions that need to be harmonized, such as the writing of CD63, CD81, Alix of Figure 2D, Figure S4 and its figure legend; the first letter of ‘k562’ in line 368 should be capitalized; words of Latin origin such as in vivo in line 449 should be italicized.

A7) The expressions were corrected as requested by reviewer.

Q8) In Figure S5, the ‘AuNP@PEG treated K562 exos’ in figure legend is mislabeled as ‘Exos c/ AuNP@PEG’ in the figure.

A8) We thank the reviewer’s careful revision. The name in figure legend was altered to “AuNP@PEG treated K562 exos”

Round 2

Reviewer 2 Report

All my concerns were addressed in this version.

Reviewer 3 Report

The authors have solved all my problems.